# CROSS-ENTROPY LOSS LEADS TO POOR MARGINS

## ABSTRACT

Neural networks could misclassify inputs that are slightly different from their training data, which indicates a small margin between their decision boundaries and the training dataset. In this work, we study the binary classification of linearly separable datasets and show that linear classifiers could also have decision boundaries that lie close to their training dataset if cross-entropy loss is used for training. In particular, we show that if the features of the training dataset lie in a low-dimensional affine subspace and the cross-entropy loss is minimized by using a gradient method, the margin between the training points and the decision boundary could be much smaller than the optimal value. This result is contrary to the conclusions of recent related works such as (Soudry et al., 2018), and we identify the reason for this contradiction. In order to improve the margin, we introduce differential training, which is a training paradigm that uses a loss function defined on pairs of points from each class. We show that the decision boundary of a linear classifier trained with differential training indeed achieves the maximum margin. The results reveal the use of cross-entropy loss as one of the hidden culprits of adversarial examples and introduces a new direction to make neural networks robust against them.

## 1 INTRODUCTION

Training neural networks is challenging and involves making several design choices. Among these are the architecture of the network, the training loss function, the optimization algorithm used for training, and their hyperparameters, such as the learning rate and the batch size. Most of these design choices influence the solution obtained by the training procedure and have been studied in detail (Kingma & Ba, 2014; Hardt et al., 2015; He et al., 2016; Wilson et al., 2017; Nar & Sastry, 2018; Smith et al., 2018). Nevertheless, one choice has been mostly taken for granted when the network is trained for a classification task: the training loss function.

Cross-entropy loss function is almost the sole choice for classification tasks in practice. Its prevalent use is backed theoretically by its association with the minimization of the Kullback-Leibler divergence between the empirical distribution of a dataset and the *confidence* of the classifier for that dataset. Given the particular success of neural networks for classification tasks (Krizhevsky et al., 2012; Simonyan & Zisserman, 2014; He et al., 2016), there seems to be little motivation to search for alternatives for this loss function, and most of the software developed for neural networks incorporates an efficient implementation for it, thereby facilitating its use.

Recently there has been a line of work analyzing the dynamics of training a linear classifier with the cross-entropy loss function (Soudry et al., 2018; Nacson et al., 2018a;b; Ji & Telgarsky, 2018). They specified the decision boundary that the gradient descent algorithm yields on linearly separable datasets and claimed that this solution achieves the maximum margin.[1] However, these claims were observed not to hold in the simple experiments we ran. For example, Figure 1 displays a case where the cross-entropy minimization for a linear classifier leads to a decision boundary which attains an extremely poor margin and is nearly orthogonal to the solution given by the hard-margin support vector machine (SVM).

We set out to understand this discrepancy between the claims of the previous works and our observations on the simple experiments. We can summarize **our contributions** as follows.

---

[1]The term "maximum margin" is used for $\ell_2$ norm throughout the paper.

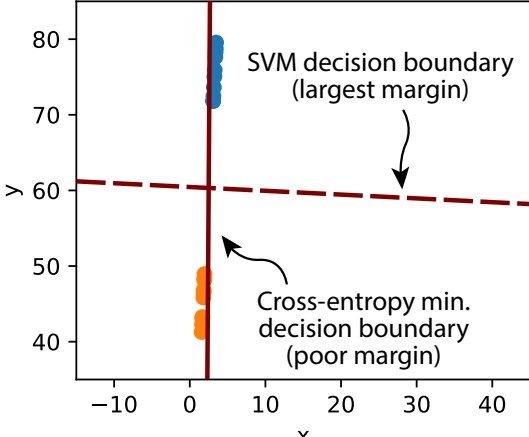

Figure 1: Orange and blue points represent the data from two different classes in $\mathbb{R}^2$. Cross-entropy minimization for a linear classifier on the given training points leads to the decision boundary shown with the solid line, which attains a very poor margin and is almost orthogonal to the solution given by the SVM.

1. We analyze the minimization of the cross-entropy loss for a linear classifier by using only two training points, i.e., only one point from each of the two classes, and we show that the dynamics of the gradient descent algorithm could yield a poor decision boundary, which could be almost orthogonal to the boundary with the maximum margin.

2. We identify the source of discrepancy between our observations and the claims of the recent works as the misleading abbreviation of notation in the previous works. We clarify why the solution obtained with cross-entropy minimization is different from the SVM solution.

3. We show that for linearly separable datasets, if the features of the training points lie in an affine subspace, and if the cross-entropy loss is minimized by a gradient method with no regularization to train a linear classifier, the margin between the decision boundary of the classifier and the training points could be much smaller than the optimal value. We verify that when a neural network is trained with the cross-entropy loss to classify two classes from the CIFAR-10 dataset, the output of the penultimate layer of the network indeed produces points that lie on an affine subspace.

4. We show that if there is no explicit and effective regularization, the weights of the last layer of a neural network could grow to infinity during training with a gradient method. Even though this has been observed in recent works as well, we are the first to point out that this divergence drives the confidence of the neural network to 100% at almost every point in the input space if the network is trained for long. In other words, the confidence depends heavily on the training duration, and its exact value might be of little significance as long as it is above 50%.

5. We introduce differential training, which is a training paradigm that uses a loss function defined on pairs of points from each class – instead of only one point from any class. We show that the decision boundary of a linear classifier trained with differential training indeed produces the SVM solution with the maximum hard margin.

## 2 CLASSIFICATION OF TWO POINTS MANIFESTS POOR MARGIN

We start with a simple binary classification problem. Given two points $x \in \mathbb{R}^d$ and $-y \in \mathbb{R}^d$ from two different classes, we can find a linear classifier by minimizing the cross-entropy loss function

$$\min_{w \in \mathbb{R}^d, b \in \mathbb{R}} \left\{ -\log\left(\frac{1}{e^{-w^\top x - b} + 1}\right) - \log\left(\frac{e^{w^\top y - b}}{e^{w^\top y - b} + 1}\right) \right\},$$

or equivalently, by solving

$$\min_{\tilde{w} \in \mathbb{R}^{d+1}} \left\{ \log(e^{-\tilde{w}^\top \tilde{x}} + 1) + \log(e^{-\tilde{w}^\top \tilde{y}} + 1) \right\}, \tag{1}$$

where $\tilde{x} = [x^\top \ 1]^\top$, $-\tilde{y} = [-y^\top \ 1]^\top$ and $\tilde{w} = [w^\top \ b]^\top$. Unless the two points $x$ and $-y$ are equal, the function (1) does not attain its minimum at a finite value of $\tilde{w}$. Consequently, if the gradient descent algorithm is used to minimize (1), the iterate at time $k$, $\tilde{w}[k]$, diverges as $k$ increases. The following theorem characterizes the growth rate of $\tilde{w}[k]$ and its direction in the limit by using a continuous-time approximation to the gradient descent algorithm.

**Theorem 1.** *Given two points $x \in \mathbb{R}^d$ and $-y \in \mathbb{R}^d$, let $\tilde{x}$ and $-\tilde{y}$ denote $[x^\top \ 1]^\top$ and $[-y^\top \ 1]$, respectively. Without loss of generality, assume $\|x\| \leq \|y\|$. If the two points are in different classes and we minimize the cross-entropy loss*

$$\min_{\tilde{w} \in \mathbb{R}^{d+1}} \ \log(1 + e^{-\tilde{w}^\top \tilde{x}}) + \log(1 + e^{-\tilde{w}^\top \tilde{y}})$$

*by using the continuous-time approximation to the gradient descent algorithm*

$$\frac{d\tilde{w}}{dt} = \tilde{x} \frac{\delta e^{-\tilde{w}^\top \tilde{x}}}{1 + e^{-\tilde{w}^\top \tilde{x}}} + \tilde{y} \frac{\delta e^{-\tilde{w}^\top \tilde{y}}}{1 + e^{-\tilde{w}^\top \tilde{y}}}$$

*with the initialization $\tilde{w}(0) = 0$ and the learning rate $\delta$, then*

$$\lim_{t \to \infty} \frac{\tilde{w}(t)}{\log(t)} = \begin{cases} \frac{\sigma_y - \sigma_{xy}}{\sigma_x \sigma_y - \sigma_{xy}^2} \tilde{x} + \frac{\sigma_x - \sigma_{xy}}{\sigma_x \sigma_y - \sigma_{xy}^2} \tilde{y} & \text{if } \sigma_{xy} < \sigma_x, \\ \frac{1}{\sigma_x} \tilde{x} & \text{if } \sigma_{xy} \geq \sigma_x, \end{cases} \tag{2}$$

*where $\sigma_x = \|\tilde{x}\|^2$, $\sigma_{xy} = \tilde{x}^\top \tilde{y}$ and $\sigma_y = \|\tilde{y}\|^2$.*

Note that first $d$ coordinates of (2) represent the normal vector of the decision boundary obtained by minimizing the cross-entropy loss (1). This vector is different from $x + y$, which is the direction of the maximum-margin solution given by the SVM. In fact, the direction in (2) could be almost orthogonal to the SVM solution in certain cases, which implies that the margin between the points and the decision boundary could be much smaller than the optimal value. Corollary 1 describes a subset of these cases.

**Corollary 1.** *Given two points $x$ and $-y$ in $\mathbb{R}^d$, let $\psi$ denote the angle between the solution given by (2) and the solution given by the SVM, i.e., $(x + y)$. If $x^\top y = 1$, then*

$$\cos^2 \psi \leq \frac{4}{2 + \frac{\sigma_y}{\sigma_x} \left(1 - \frac{1}{\sigma_x}\right)},$$

*where $\sigma_x = \|x\|^2 + 1$ and $\sigma_y = \|y\|^2 + 1$. Consequently, as $\|x\|/\|y\|$ approaches 0 while maintaining the condition $x^\top y = 1$, the angle $\psi$ converges to $\pi/2$.*

**Remark 1.** Corollary 1 shows that if $x$ and $-y$ have disparate norms, the minimization of the cross-entropy loss with gradient descent algorithm could lead to a direction which is almost orthogonal to the maximum-margin solution. It may seem like this problem could be avoided with preprocessing the data so as to normalize the data points. However, this approach will not be effective for neural networks: if we consider an $L$-layer neural network, $w^\top \phi_{L-1}(x)$, and regard the first $L - 1$ layers, $\phi_{L-1}(\cdot)$, as a feature mapping, preprocessing a dataset $\{x_i\}_{i \in I}$ will not produce a normalized set of features $\{\phi_{L-1}(x_i)\}_{i \in I}$. Note that we could not normalize $\{\phi_{L-1}(x_i)\}_{i \in I}$ directly either, since the mapping $\phi_{L-1}(\cdot)$ evolves during training.

**Remark 2.** Theorem 1 shows that the norm of $w$ keeps growing unboundedly as the training continues. The same behavior will be observed for larger datasets in the next sections as well. Since the "confidence" of the classifier for its prediction at a point $x$ is given by

$$\max \left( \frac{1}{e^{-w^\top x - b} + 1}, \frac{e^{-w^\top x - b}}{e^{-w^\top x - b} + 1} \right),$$

this unbounded growth of $\|w\|$ drives the confidence of the classifier to 100% at every point in the input space, except at the points on the decision boundary, if the algorithm is run for long. Given the lack of effective regularization for neural networks, a similar unbounded growth is expected to be observed in neural network training as well, which is mentioned in (Bartlett et al., 2017). As a result, the confidence of a neural network might be highly correlated with the training duration, and whether a neural network gives 99% or 51% confidence for a prediction might be of little importance as long as it is above 50%. In other words, regarding this confidence value as a measure of similarity between an input and the training dataset from the most-likely class should be reconsidered.

## 3 DATA WITH LOW-DIMENSIONAL FEATURES ATTAIN POOR MARGIN

In this section, we examine the binary classification of a linearly separable dataset by minimizing the cross-entropy loss function. Recently, this problem has also been studied in (Soudry et al., 2018; Nacson et al., 2018b;a; Ji & Telgarsky, 2018). We restate an edited version of the main theorem of (Soudry et al., 2018), followed by the reason of the edition.

**Theorem 2** (Adapted from Theorem 3 of (Soudry et al., 2018))**.** *Given two sets of points $\{x_i\}_{i \in I}$ and $\{-y_j\}_{j \in J}$ that are linearly separable in $\mathbb{R}^d$, let $\tilde{x}_i$ and $-\tilde{y}_j$ denote $[x_i^\top \ 1]^\top$ and $[-y_j^\top \ 1]^\top$, respectively, for all $i \in I$, $j \in J$. Then the iterate of the gradient descent algorithm, $\tilde{w}(t)$, on the cross-entropy loss function*

$$\min_{\tilde{w} \in \mathbb{R}^{d+1}} \sum_{i \in I} \log(1 + e^{-\tilde{w}^\top \tilde{x}_i}) + \sum_{j \in J} \log(1 + e^{-\tilde{w}^\top \tilde{y}_j}) \tag{3}$$

*with a sufficiently small step size will converge in direction:*

$$\lim_{t \to \infty} \frac{\tilde{w}(t)}{\|\tilde{w}(t)\|} = \frac{\overline{w}}{\|\overline{w}\|},$$

*where $\overline{w}$ is the solution to*

$$\overline{w} = \operatorname*{argmin}_{u \in \mathbb{R}^{d+1}} \|u\|^2 \quad s.t. \ \langle u, \tilde{x}_i \rangle \geq 1, \ \langle u, \tilde{y}_j \rangle \geq 1 \quad \forall i \in I, \forall j \in J. \tag{4}$$

The solution (4) given in Theorem 2 was referred in (Soudry et al., 2018), and consequently in the other works, as the maximum-margin solution. However, due to the misleading absence of the bias term in the notation, this is incorrect. Given the linearly separable sets of points $\{x_i\}_{i \in I}$ and $\{-y_j\}_{j \in J}$, the maximum-margin solution given by the SVM solves

$$\begin{aligned} \underset{w, b}{\text{minimize}} \quad & \|w\|_2^2 \\ \text{subject to} \quad & \langle w, x_i \rangle + b \geq 1 \qquad \forall i \in I, \\ & \langle w, -y_j \rangle + b \leq -1 \quad \forall j \in J. \end{aligned} \tag{P1}$$

On the other hand, the solution given by Theorem 2 corresponds to

$$\begin{aligned} \underset{w, b}{\text{minimize}} \quad & \|w\|_2^2 + b^2 \\ \text{subject to} \quad & \langle w, x_i \rangle + b = \langle \tilde{w}, \tilde{x}_i \rangle \geq 1 \qquad \forall i \in I, \\ & \langle w, -y_j \rangle + b = \langle \tilde{w}, -\tilde{y}_j \rangle \leq -1 \quad \forall j \in J, \end{aligned} \tag{P2}$$

where we define $\tilde{w} = [w^\top \ b]^\top$, $\tilde{x}_i = [x_i^\top \ 1]^\top$ and $\tilde{y}_j = [y_j^\top \ -1]^\top$ for all $i \in I, j \in J$. Even though the sets of constraints for both problems are identical, their objective functions are different, and consequently, the solutions are different. As a result, the decision boundary obtained by cross-entropy minimization does not necessarily attain the maximum hard margin. In fact, as the following theorem shows, its margin could be arbitrarily worse than the maximum margin.

**Theorem 3.** *Assume that the points $\{x_i\}_{i \in I}$ and $\{-y_j\}_{j \in J}$ are linearly separable and lie in an affine subspace; that is, there exist a set of orthonormal vectors $\{r_k\}_{k \in K}$ and a set of scalars $\{\Delta_k\}_{k \in K}$ such that*

$$\langle r_k, x_i \rangle = \langle r_k, -y_j \rangle = \Delta_k \quad \forall i \in I, \forall j \in J, \forall k \in K.$$

*Let $\langle \overline{w}, \cdot \rangle + B = 0$ denote the decision boundary obtained by minimizing the cross-entropy loss, i.e. the pair $(\overline{w}, B)$ solves*

$$\min_{w, b} \|w\|^2 + b^2 \ s.t. \ \langle w, x_i \rangle + b \geq 1, \ \langle w, -y_j \rangle + b \leq -1 \quad \forall i \in I, \ \forall j \in J.$$

*Then the minimization of the cross-entropy loss (3) yields a margin smaller than or equal to*

$$\frac{1}{\sqrt{\frac{1}{\gamma^2} + B^2 \sum_{k \in K} \Delta_k^2}}$$

*where $\gamma$ denotes the optimal hard margin given by the SVM solution.*

**Remark 3.** Theorem 3 shows that if the training points lie in an affine subspace, the margin obtained by the cross-entropy minimization will be smaller than the optimal margin value. As the dimension of this affine subspace decreases, the cardinality of the set $K$ increases and the term $\sum_{k \in K} \Delta_k^2$ could become much larger than $1/\gamma^2$. Therefore, as the dimension of the subspace containing the training points gets smaller compared to the dimension of the input space, cross-entropy minimization with a gradient method becomes more likely to yield a poor margin. Note that this argument also holds for classifiers of the form $w^\top \phi(x)$ with the fixed feature mapping $\phi(\cdot)$.

The next theorem relaxes the condition of Theorem 3 and allows the training points to be near an affine subspace instead of being exactly on it. Note that the ability to compare the margin obtained by cross-entropy minimization with the optimal value is lost. Nevertheless, it highlights the fact that same set of points could be assigned a different margin by cross-entropy minimization if all of them are shifted away from the origin by the same amount in the same direction.

**Theorem 4.** *Assume that the points $\{x_i\}_{i \in I}$ and $\{-y_j\}_{j \in J}$ in $\mathbb{R}^d$ are linearly separable and there exist a set of orthonormal vectors $\{r_k\}_{k \in K}$ and a set of scalars $\{\Delta_k\}_{k \in K}$ such that*

$$\langle r_k, x_i \rangle \geq \Delta_k, \ \langle r_k, -y_j \rangle \leq \Delta_k \quad \forall i \in I, \ \forall j \in J, \ \forall k \in K.$$

*Let $\langle \overline{w}, \cdot \rangle + B = 0$ denote the decision boundary obtained by minimizing the cross-entropy loss, i.e. the pair $(\overline{w}, B)$ solves*

$$\min_{w,b} \ \|w\|^2 + b^2 \ \text{ s.t. } \langle w, x_i \rangle + b \geq 1, \ \langle w, -y_j \rangle + b \leq -1 \quad \forall i \in I, \ \forall j \in J.$$

*Then the minimization of the cross-entropy loss (3) yields a margin smaller than or equal to*

$$\frac{1}{\sqrt{B^2 \sum_{k \in K} \Delta_k^2}}$$

**Remark 4.** Both Theorem 3 and Theorem 4 consider linearly separable datasets. If the dataset is not linearly separable, (Ji & Telgarsky, 2018) predicts that the normal vector of the decision boundary, $w$, will have two components, one of which converges to a finite vector and the other diverges. The diverging component still has the potential to drive the decision boundary to a direction with a poor margin. In fact, the margin is expected to be small especially if the points intruding into the opposite class lie in the same subspace as the optimal normal vector for the decision boundary. In this work, we focus on the case of separable datasets as this case provides critical insight into the issues of state-of-the-art neural networks, given they can easily attain zero training error even on randomly generated datasets, which indicates the linear separability of the features obtained at their penultimate layers (Zhang et al., 2017).

## 4 Differential Training Improves Margin

In previous sections, we saw that the cross-entropy minimization could lead to poor margins, and the main reason for this was the appearance of the bias term in the objective function of (P2). In order to remove the effect of the bias term, consider the SVM problem (P1) and note that this problem could be equivalently written as

$$\begin{aligned} &\underset{w}{\text{minimize}} && \|w\|_2^2 \\ &\text{subject to} && \langle w, x_i + y_j \rangle \geq 2 \quad \forall i \in I, \ \forall j \in J \end{aligned} \tag{P3}$$

if we only care about the weight parameter $w$. This gives the hint that if we use the set of differences $\{x_i + y_j : i \in I, j \in J\}$ instead of the individual sets $\{x_i\}_{i \in I}$ and $\{-y_j\}_{j \in J}$, the bias term could be excluded from the problem. This was also noted in (Keerthi et al., 2000; Ishibashi et al., 2008) previously. Indeed, this approach allows obtaining the SVM solution with a loss function similar to the cross-entropy loss, as the following theorem shows.

**Theorem 5.** *Given two sets of points $\{x_i\}_{i \in I}$ and $\{-y_j\}_{j \in J}$ that are linearly separable in $\mathbb{R}^d$, if we solve*

$$\min_{w \in \mathbb{R}^d} \sum_{i \in I} \sum_{j \in J} \log(1 + e^{-w^\top (x_i + y_j)}) \tag{5}$$

*by using the gradient descent algorithm with a sufficiently small learning rate, the direction of $w$ converges to the direction of maximum-margin solution, i.e.*

$$\lim_{t\to\infty} \frac{w(t)}{\|w(t)\|} = \frac{w_{SVM}}{\|w_{SVM}\|}, \tag{6}$$

*where $w_{SVM}$ is the solution of (P3).*

*Proof.* Apply Theorem 2 by replacing the sets $\{x_i\}_{i\in I}$ and $\{-y_j\}_{j\in J}$ with $\{x_i + y_j\}_{i\in I, j\in J}$ and the empty set, respectively. Then the minimization of the loss function (5) with the gradient descent algorithm leads to

$$\lim_{t\to\infty} \frac{w}{\|w\|} = \frac{\overline{w}}{\|\overline{w}\|}$$

where $\overline{w}$ satisfies

$$\overline{w} = \arg\min_{w} \|w\|^2 \text{ such that } \langle w, x_i + y_j\rangle \geq 1 \quad \forall i \in I, \forall j \in J.$$

Since $w_{\text{SVM}}$ is the solution of (P3), we obtain $\overline{w} = \frac{1}{2} w_{\text{SVM}}$, and the claim of the theorem holds. $\blacksquare$

**Remark 5.** Theorem 5 is stated for the gradient descent algorithm, but the identical statement could be made for the stochastic gradient method as well by invoking the main theorem of (Nacson et al., 2018).

Minimization of the cost function (5) yields the weight parameter $\hat{w}$ of the decision boundary. The bias parameter, $b$, could be chosen by plotting the histogram of the inner products $\{\langle \hat{w}, x_i\rangle\}_{i\in I}$ and $\{\langle \hat{w}, -y_j\rangle\}_{j\in J}$ and fixing a value for $\hat{b}$ such that

$$\langle \hat{w}, x_i\rangle + \hat{b} \geq 0 \quad \forall i \in I, \tag{7a}$$

$$\langle \hat{w}, -y_j\rangle + \hat{b} \leq 0 \quad \forall j \in J. \tag{7b}$$

The largest hard margin is achieved by

$$\hat{b} = -\frac{1}{2}\min_{i\in I}\langle \hat{w}, x_i\rangle - \frac{1}{2}\max_{j\in J}\langle \hat{w}, -y_j\rangle. \tag{8}$$

However, by choosing a larger or smaller value for $\hat{b}$, it is possible to make a tradeoff between the Type-I and Type-II errors.

The cost function (5) includes a loss defined on every pair of data points from the two classes. This cost function can be considered as the cross-entropy loss on a new dataset which contains $|I| \times |J|$ points. There are two aspects of this fact:

1. When standard loss functions are used for classification tasks, we need to oversample or undersample either of the classes if the training dataset contains different number of points from different classes. This problem does not arise when we use the cost function (5).

2. Number of pairs in the new dataset, $|I| \times |J|$, will usually be much larger than the original dataset, which contains $|I| + |J|$ points. Therefore, the minimization of (5) might appear more expensive than the minimization of the standard cross-entropy loss computationally. However, if the points in different classes are well separated and the stochastic gradient method is used to minimize (5), the algorithm achieves zero training error after using only a few pairs, which is formalized in Theorem 6. Further computation is needed only to improve the margin of the classifier. In addition, in our experiments to train a neural network to classify two classes from the CIFAR-10 dataset, only a few percent of $|I| \times |J|$ points were observed to be sufficient to reach a high accuracy on the training dataset.

**Theorem 6.** *Given two sets of points $\{x_i\}_{i\in I}$ and $\{-y_j\}_{j\in J}$ that are linearly separable in $\mathbb{R}^d$, assume the cost function (5) is minimized with the stochastic gradient method. Define*

$$R_x = \max\{\|x_i - x_{i'}\| : i, i' \in I\}, \quad R_y = \max\{\|y_j - y_{j'}\| : j, j' \in J\}$$

*and let $\gamma$ denote the hard margin that would be obtained with the SVM:*

$$2\gamma = \max_{u\in\mathbb{R}^d} \min_{i\in I, j\in J} \langle x_i + y_j, u/\|u\|\rangle.$$

*If $2\gamma \geq 5\max(R_x, R_y)$, then the stochastic gradient algorithm produces a weight parameter, $\hat{w}$, only in one iteration which satisfies the inequalities (7a)-(7b) along with the bias, $\hat{b}$, given by (8).*

## 5 NUMERICAL EXPERIMENTS

In this section, we present numerical experiments supporting our claims.

**Differential training.** In Figure 2, we show the decision boundaries of two linear classifiers, where one of them is trained by minimizing the *cross-entropy loss*, and the other through *differential training*. Unlike the example shown in Figure 1, here the data do not exactly lie in an affine subspace. In particular, one of the classes is composed of 10 samples from a normal distribution with mean $(2, 12)$ and variance 25, and the other class is composed of 10 samples from a normal distribution with mean $(40, 50)$ and variance 25. As can be seen from the figure, the cross-entropy minimization yields a margin that is smaller than differential training, even though when the training dataset is not low-dimensional, which is predicted by Theorem 4.

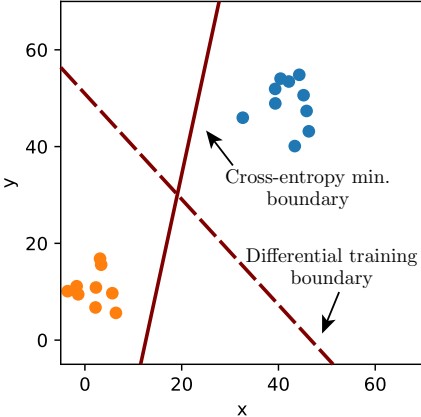

Figure 2: Classification boundaries obtained using *differential training* and *cross-entropy minimization*. The margin recovered by cross-entropy minimization is worse than differential training even when the training dataset is not low-dimensional.

**Low-dimensionality.** We empirically evaluated if the features obtained at the penultimate layer of a neural network indeed lie in a low-dimensional affine subspace. For this purpose, we trained a convolutional neural network architecture to classify horses and planes from the CIFAR-10 dataset (Krizhevsky & Hinton, 2009). Figure 3 shows the cumulative variance explained for the features that feed into the soft-max layer as a function of the number of principle components used. We observe that the features, which are the outputs of the penultimate layer of the network, lie in a low-dimensional affine subspace, and this holds for a variety of training modalities for the network. This observation is relevant to Remark 3. The dimension of the subspace containing the training points is at most 20, which is much smaller than the dimension of the feature space, 84. Consequently, cross-entropy minimization with a gradient method is expected to yield a poor margin on these features.

## 6 DISCUSSION

We compare our results with related works and discuss their implications for the following subjects.

**Adversarial examples.** State-of-the-art neural networks have been observed to misclassify inputs that are slightly different from their training data, which indicates a small margin between their decision boundaries and the training dataset (Szegedy et al., 2013; Goodfellow et al., 2015; Moosavi-Dezfooli et al., 2017; Fawzi et al., 2017). Our results reveal that the combination of gradient methods, cross-entropy loss function and the low-dimensionality of the training dataset (at least in some domain) has a responsibility for this problem. Note that SVM with the radial basis function was shown to be robust against adversarial examples, and this was attributed to the high nonlinearity of the radial basis function in (Goodfellow et al., 2015). Given that the SVM uses neither the cross entropy loss function nor the gradient descent algorithm for training, we argue that the robustness of SVM is no surprise – independent of its nonlinearity. Lastly, effectiveness of differential training for neural networks against adversarial examples is our ongoing work.

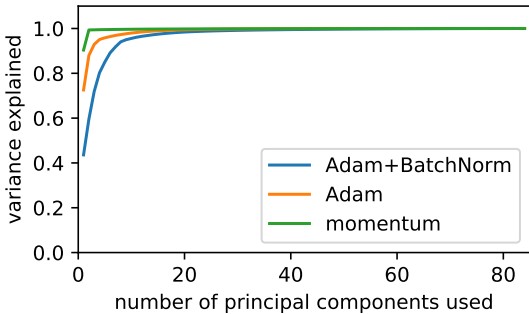

Figure 3: The activations feeding into the soft-max layer could be considered as the features for a linear classifier. Plot shows the cumulative variance explained for these features as a function of the number of principle components used. Almost all the variance in the features is captured by the first 20 principle components out of 84, which shows that the input to the soft-max layer resides predominantly in a low-dimensional subspace.

**Low-dimensionality of the training dataset.** As stated in Remark 3, as the dimension of the affine subspace containing the training dataset gets very small compared to the dimension of the input space, the training algorithm will become more likely to yield a small margin for the classifier. This observation confirms the results of (Marzi et al., 2018), which showed that if the set of training data is projected onto a low-dimensional subspace before feeding into a neural network, the performance of the network against adversarial examples is improved – since projecting the inputs onto a low-dimensional domain corresponds to decreasing the dimension of the input space. Even though this method is effective, it requires the knowledge of the domain in which the training points are low-dimensional. Because this knowledge will not always be available, finding alternative training algorithms and loss functions that are suited for low-dimensional data is still an important direction for future research.

**Robust optimization.** Using robust optimization techniques to train neural networks has been shown to be effective against adversarial examples (Madry et al., 2018; Athalye et al., 2018). Note that these techniques could be considered as inflating the training points by a presumed amount and training the classifier with these inflated points. Consequently, as long as the cross-entropy loss is involved, the decision boundaries of the neural network will still be in the vicinity of the inflated points. Therefore, even though the classifier is robust against the disturbances of the presumed magnitude, the margin of the classifier could still be much smaller than what it could potentially be.

**Differential training.** We introduced differential training, which allows the feature mapping to remain trainable while ensuring a large margin between different classes of points. Therefore, this method combines the benefits of neural networks with those of support vector machines. Even though moving from $2N$ training points to $N^2$ seems prohibitive, it points out that a true classification should in fact be able to differentiate between the pairs that are hardest to differentiate, and this search will necessarily require an $N^2$ term. Some heuristic methods are likely to be effective, such as considering only a smaller subset of points closer to the boundary and updating this set of points as needed during training. If a neural network is trained with this procedure, the network will be forced to find features that are able to tell apart between the hardest pairs.

**Nonseparable data.** What happens when the training data is not linearly separable is an open direction for future work. However, as stated in Remark 4, this case is not expected to arise for the state-of-the-art networks, since they have been shown to achieve zero training error even on randomly generated datasets (Zhang et al., 2017), which implies that the features represented by the output of their penultimate layer eventually become linearly separable.

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

# A    PROOF OF THEOREM 1

Theorem 1 could be proved by using Theorem 2, but we provide an independent proof here. Gradient descent algorithm with learning rate $\delta$ on the cross-entropy loss (1) yields

$$\frac{d\tilde{w}}{dt} = \delta\tilde{x}\frac{e^{-\tilde{w}^\top\tilde{x}}}{1 + e^{-\tilde{w}^\top\tilde{x}}} + \delta\tilde{y}\frac{e^{-\tilde{w}^\top\tilde{y}}}{1 + e^{-\tilde{w}^\top\tilde{y}}}.$$

If $\tilde{w}(0) = 0$, then $\tilde{w}(t) = p(t)\tilde{x} + q(t)\tilde{y}$ for all $t \geq 0$, where

$$\dot{p} = \delta\frac{e^{-p\|\tilde{x}\|^2 - q\langle\tilde{x},\tilde{y}\rangle}}{1 + e^{-p\|\tilde{x}\|^2 - q\langle\tilde{x},\tilde{y}\rangle}}, \quad \dot{q} = \delta\frac{e^{-q\|\tilde{y}\|^2 - p\langle\tilde{x},\tilde{y}\rangle}}{1 + e^{-q\|\tilde{y}\|^2 - p\langle\tilde{x},\tilde{y}\rangle}}.$$

Define

$$\alpha = p\|\tilde{x}\|^2 + q\langle\tilde{x},\tilde{y}\rangle, \quad \beta = q\|\tilde{y}\|^2 + p\langle\tilde{x},\tilde{y}\rangle,$$

$$a = \delta\|\tilde{x}\|^2 = \delta\sigma_x, \quad c = \delta\|\tilde{y}\|^2 = \delta\sigma_y, \quad b = \delta\langle\tilde{x},\tilde{y}\rangle = \delta\sigma_{xy}.$$

Then we can write

$$\dot{\alpha} = a\frac{e^{-\alpha}}{1 + e^{-\alpha}} + b\frac{e^{-\beta}}{1 + e^{-\beta}},$$

$$\dot{\beta} = c\frac{e^{-\beta}}{1 + e^{-\beta}} + b\frac{e^{-\alpha}}{1 + e^{-\alpha}}.$$

Finally, define $z = e^\alpha$ and $v = e^\beta$ so that

$$\dot{z} = \frac{z}{z + 1}\left(a + b\frac{z + 1}{v + 1}\right),$$

$$\dot{v} = \frac{v}{v + 1}\left(c + b\frac{v + 1}{z + 1}\right).$$

Without loss of generality, assume $c \geq a$. Before we proceed, note that

$$\frac{d}{dt}\left(\frac{z}{v}\right) = \frac{c - b}{z + 1}\frac{z}{v}\left(\frac{a - b}{c - b} - \frac{z + 1}{v + 1}\right),$$

$$\frac{d}{dt}\left(\frac{z + 1}{v + 1}\right) = \frac{z}{(v + 1)^2}\left[\frac{v + 1}{z + 1}a - \frac{v}{z}b - \left(\frac{v}{z}\frac{z + 1}{v + 1}c - b\right)\right].$$

Let $u$ and $w$ denote $\frac{z}{v}$ and $\frac{z+1}{v+1}$, respectively. Then,

$$\dot{u} < 0 \quad \text{if } w > \frac{a - b}{c - b}$$

$$\dot{u} > 0 \quad \text{if } w < \frac{a - b}{c - b}$$

$$\dot{w} < 0 \quad \text{if } \left(\frac{a}{w} + b\right)u < cw + b$$

$$\dot{w} > 0 \quad \text{if } \left(\frac{a}{w} + b\right)u > cw + b$$

**Lemma 1.** *If $b = 0$, then*

$$\lim_{t\to\infty}\frac{\tilde{w}(t)}{\log(t)} = \frac{1}{\|\tilde{x}\|}\tilde{x} + \frac{1}{\|\tilde{y}\|}\tilde{y}.$$

*Proof.* Note that

$$\frac{d(z + \log(z))}{dt} = a \implies z(t) - z_0 + \log(z(t)/z_0) = at,$$

$$\frac{d(v + \log(v))}{dt} = c \implies v(t) - v_0 + \log(v(t)/v_0) = ct.$$

Then,

$$\lim_{t\to\infty}\frac{\alpha(t)}{\log(t)} = \lim_{t\to\infty}\frac{\log(z(t))}{\log(t)} = 1 = \lim_{t\to\infty}\frac{\log(v(t))}{\log(t)} = \lim_{t\to\infty}\frac{\beta(t)}{\log(t)},$$

and

$$\lim_{t\to\infty}\frac{\tilde{w}(t)}{\log(t)} = \frac{\tilde{x}}{\|\tilde{x}\|^2} + \frac{\tilde{y}}{\|\tilde{y}\|^2}. \qquad\blacksquare$$

**Lemma 2.** *If $b < 0$, then there exists $t_0 \in (0,\infty)$ such that*

$$\frac{-b}{c} \le \frac{z+1}{v+1} \le \frac{a}{-b} \quad \forall t \ge t_0.$$

*Proof.* Note that $\frac{-b}{c} \le \frac{a-b}{c-b} \le \frac{a}{-b}$ because $b < 0$. First assume $\frac{z_0+1}{v_0+1} \ge \frac{a}{-b}$. Then, $\dot{z} \le 0$ and

$$\dot{v} = \frac{v}{v+1}\left(c + b\frac{v+1}{z+1}\right) \ge \frac{v}{v+1}\left(c - \frac{b^2}{a}\right) \ge \frac{v_0}{v_0+1}\frac{ac-b^2}{a},$$

which implies that

$$\frac{z+1}{v+1} \le (z_0+1)\left(\frac{v_0}{v_0+1}\frac{ac-b^2}{a}t + 1\right)^{-1}$$

as long as $\frac{z+1}{v+1} \ge \frac{a}{-b}$, and this can be satisfied only for a finite time. Now assume $\frac{z_0+1}{v_0+1} \le \frac{-b}{c}$. Then, $\dot{v} \le 0$ and

$$\dot{z} = \frac{z}{z+1}\left(a + b\frac{z+1}{v+1}\right) \ge \frac{z_0}{z_0+1}\frac{ac-b^2}{c},$$

which implies

$$\frac{z+1}{v+1} \ge \left(\frac{z_0}{z_0+1}\frac{ac-b^2}{c}t + 1\right)(v_0+1)^{-1}$$

as long as $\frac{z+1}{v+1} \le \frac{-b}{c}$, and this can be satisfied only for a finite time as well. $\qquad\blacksquare$

**Lemma 3.** *If $b < 0$, then*

$$0 \le \dot{z} \le \frac{ac-b^2}{c}, \quad 0 \le \dot{v} \le \frac{ac-b^2}{a} \quad \forall t \ge t_0,$$

*where $t_0$ is given by Lemma 2.*

*Proof.*

$$\dot{z} = \frac{z}{z+1}\left(a + b\frac{z+1}{v+1}\right) \le \frac{z}{z+1}\frac{ac-b^2}{c} \le \frac{ac-b^2}{c}$$

$$\dot{v} = \frac{v}{v+1}\left(c + b\frac{v+1}{z+1}\right) \le \frac{v}{v+1}\frac{ac-b^2}{a} \le \frac{ac-b^2}{a} \qquad\blacksquare$$

**Lemma 4.** *If $b < 0$, then*

$$\lim_{t\to\infty}\frac{\log(z)}{\log t} = \lim_{t\to\infty}\frac{\log(v)}{\log t} = 1$$

*Proof.* From Lemma 3,

$$\dot{v} \le 0 \iff c + b\frac{v+1}{z+1} \ge 0,$$

and

$$v \le \frac{ac-b^2}{a}t + v_0'.$$

Combining these two inequalities, we have

$$c + \frac{b}{a}\left[(ac-b^2)t + v_0' + 1\right]\frac{1}{z+1} \ge 0 \iff z+1 \ge \frac{-b}{ac}[(ac-b^2)t + v_0' + 1]$$

As a result,

$$\frac{(-b)(ac - b^2)}{ac}t + z_1 \leq z(t) \leq \frac{ac - b^2}{c}t + z_2 \quad \forall t \geq t_0 \implies \lim_{t \to \infty} \frac{\log(z)}{\log(t)} = 1.$$

By using Lemma 2,

$$\log(-b/c) \leq \log(z + 1) - \log(v + 1) \leq \log(-a/b) \implies \lim_{t \to \infty} \frac{\log(v)}{\log(t)} = 1. \qquad \blacksquare$$

**Lemma 5.** *If $b < 0$, then*

$$\lim_{t \to \infty} \frac{\tilde{w}(t)}{\log(t)} = \delta \frac{c - b}{ac - b^2}\tilde{x} + \delta \frac{a - b}{ac - b^2}\tilde{y} = \frac{\sigma_y - \sigma_{xy}}{\sigma_x \sigma_y - \sigma_{xy}^2}\tilde{x} + \frac{\sigma_x - \sigma_{xy}}{\sigma_x \sigma_y - \sigma_{xy}^2}\tilde{y}$$

*Proof.* Solving the set of equations

$$1 = \lim \frac{\alpha}{\log(t)} = \frac{a}{\delta}\lim \frac{p}{\log(t)} + \frac{b}{\delta}\lim \frac{q}{\log(t)},$$

$$1 = \lim \frac{\beta}{\log(t)} = \frac{c}{\delta}\lim \frac{q}{\log(t)} + \frac{b}{\delta}\lim \frac{p}{\log(t)},$$

we obtain

$$\lim_{t \to \infty} \frac{p}{\log(t)} = \delta \frac{c - b}{ac - b^2}, \quad \lim_{t \to \infty} \frac{q}{\log(t)} = \delta \frac{a - b}{ac - b^2} \qquad \blacksquare$$

**Lemma 6.** *If $b > 0$, then*

$$\lim_{t \to \infty} \frac{z}{v} = \lim_{t \to \infty} \frac{z + 1}{v + 1} = \begin{cases} 0 & \text{if } a \leq b \\ \frac{a - b}{c - b} & \text{if } a > b \end{cases}$$

*Proof.* Note that $\dot{z} \geq a/2$ and $\dot{v} \geq c/2$; therefore,

$$\lim_{t \to \infty} \frac{z + 1}{v + 1} = \lim_{t \to \infty} \frac{z}{v} \implies \lim_{t \to \infty} u = \lim_{t \to \infty} w$$

if either side exists. Remember that

$$\dot{w} < 0 \iff u < \frac{cw^2 + bw}{a + bw} =: f(w).$$

We can compute

$$f'(w) = \frac{2acw + bcw^2 + ab}{b^2 w^2 + 2abw + a^2}.$$

The function $f$ is strictly increasing and convex for $w > 0$. We have

$$f(0) = 0,$$

$$f\left(\frac{a - b}{c - b}\right) = \frac{a - b}{c - b}.$$

Therefore, when $b \geq a$, the only fixed point of $f$ over $[0, \infty)$ is the origin, and when $a > b$, 0 and $(a - b)/(c - b)$ are the only fixed points of $f$ over $[0, \infty)$.

Figure 4 shows the curves over which $\dot{u} = 0$ and $\dot{w} = 0$. Since $\lim_{t \to \infty} u = \lim_{t \to \infty} w$, the only points $(u, w)$ can converge to are the fixed points of $f$. Remember that

$$\dot{u} = \frac{c - b}{z + 1}u\left(\frac{a - b}{c - b} - w\right),$$

so when $a > b$, the origin $(0, 0)$ is unstable in the sense of Lyapunov, and $(u, w)$ cannot converge to it. Otherwise, $(0, 0)$ is the only fixed point, and it is stable. As a result,

$$\lim_{t \to \infty} \frac{z}{v} = \lim_{t \to \infty} \frac{z + 1}{v + 1} = \begin{cases} 0 & \text{if } a \leq b \\ \frac{a - b}{c - b} & \text{if } a > b \end{cases} \qquad \blacksquare$$

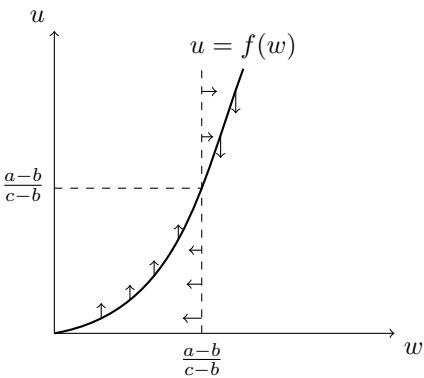

Figure 4: Stationary points of function $f$.

**Lemma 7.** *If $a > b > 0$, then*

$$\lim_{t \to \infty} \frac{\tilde{w}(t)}{\log(t)} = \delta \frac{c - b}{ac - b^2} \tilde{x} + \delta \frac{a - b}{ac - b^2} \tilde{y} = \frac{\sigma_y - \sigma_{xy}}{\sigma_x \sigma_y - \sigma_{xy}^2} \tilde{x} + \frac{\sigma_x - \sigma_{xy}}{\sigma_x \sigma_y - \sigma_{xy}^2} \tilde{y}.$$

*Proof.* From Lemma 6,

$$\lim_{t \to \infty} \frac{z}{t} = \lim_{t \to \infty} \dot{z} = \lim_{t \to \infty} \frac{z}{z + 1} \left( a + b \frac{z + 1}{v + 1} \right) = a + b \frac{a - b}{c - b} = \frac{ac - b^2}{c - b},$$

$$\lim_{t \to \infty} \frac{v}{t} = \frac{ac - b^2}{a - b}.$$

Consequently,

$$\lim_{t \to \infty} \frac{\log(z)}{\log(t)} = \lim_{t \to \infty} \frac{\log(v)}{\log(t)} = 1$$

which gives the same solution as Lemma 5:

$$\lim_{t \to \infty} \frac{p}{\log(t)} = \delta \frac{c - b}{ac - b^2}, \quad \lim_{t \to \infty} \frac{q}{\log(t)} = \delta \frac{a - b}{ac - b^2}. \qquad \blacksquare$$

**Lemma 8.** *If $b \geq a$, then*

$$\lim_{t \to \infty} \frac{\tilde{w}(t)}{\log(t)} = \frac{1}{\|\tilde{x}\|} \tilde{x}$$

*Proof.*

$$\lim_{t \to \infty} \frac{z}{t} = a, \quad \lim_{t \to \infty} \frac{\log(z)}{\log(t)} = 1,$$

$$\lim_{t \to \infty} \frac{v}{t} = \lim_{t \to \infty} \dot{v} = \infty$$

$$\lim_{t \to \infty} \frac{\log(v)}{\log(t)} = \lim_{t \to \infty} \frac{\dot{v}}{v} t = \lim_{t \to \infty} \frac{1}{v + 1} \left( c + b \frac{v + 1}{z + 1} \right) t = \lim_{t \to \infty} \frac{ct}{v + 1} + \lim_{t \to \infty} \frac{bt}{z + 1} = \frac{b}{a}$$

$$\lim_{t \to \infty} \frac{p}{\log(t)} = \frac{1}{\|\tilde{x}\|^2}, \quad \lim_{t \to \infty} \frac{q}{\log(t)} = 0 \implies \lim_{t \to \infty} \frac{px + qy}{\log(t)} = \frac{1}{\|\tilde{x}\|^2} \tilde{x} \qquad \blacksquare$$

**Proof of Theorem 1.** Lemma 1, Lemma 5, Lemma 7 and Lemma 8 prove all cases of Theorem 1. $\blacksquare$

# B    PROOF OF COROLLARY 1

Since $x^\top y = 1$, we have $\sigma_{xy} = \tilde{x}^\top \tilde{y} = x^\top y - 1 = 0 < a$. Then the normal vector of the decision boundary is proportional to $\frac{1}{\sigma_x} x + \frac{1}{\sigma_y} y$. For the angle between this vector and the solution of the SVM, we can write

$$\cos \psi = \frac{\langle \frac{1}{\sigma_x} x + \frac{1}{\sigma_y} y, x + y \rangle}{\left\| \frac{1}{\sigma_x} x + \frac{1}{\sigma_y} y \right\| \|x + y\|} = \frac{2}{\left\| \frac{1}{\sigma_x} x + \frac{1}{\sigma_y} y \right\| \sqrt{\sigma_x + \sigma_y}}.$$

We can obtain a lower bound for square of the denominator as

$$\left\| \frac{1}{\sigma_x} x + \frac{1}{\sigma_y} y \right\|^2 (\sigma_x + \sigma_y) \geq \left( 2 + \frac{\sigma_y}{\sigma_x} - \frac{\sigma_y}{\sigma_x^2} \right) + \left( \frac{1}{\sigma_x} + \frac{1}{\sigma_y} + \frac{\sigma_x}{\sigma_y} - \frac{\sigma_x}{\sigma_y^2} \right) \geq 2 + \frac{\sigma_y}{\sigma_x} \left( 1 - \frac{1}{\sigma_x} \right).$$

As a result,

$$\cos^2 \psi \leq \frac{4}{2 + \frac{\sigma_y}{\sigma_x} \left( 1 - \frac{1}{\sigma_x} \right)}. \qquad \blacksquare$$

# C    PROOF OF THEOREM 3

Let $\langle w_{\text{SVM}}, \cdot \rangle + b_{\text{SVM}} = 0$ denote the hyperplane obtained as the solution of SVM, i.e., $(w_{\text{SVM}}, b_{\text{SVM}})$ is the solution to the problem

$$\min_{w,b} \|w\|^2 \ \text{s.t.} \ \langle w, x_i \rangle + b \geq 1, \ \langle w, -y_j \rangle + b \leq -1 \ \forall i \in I, \ \forall j \in J.$$

Assume that $\overline{w} = u + \sum_{k=1}^m \alpha_k r_k$, where $u \in \mathbb{R}^d$ and $\langle u, r_k \rangle = 0$ for all $k \in K$.

The Lagrangian of the problem (4) can be written as

$$\frac{1}{2} \|w\|^2 + \frac{1}{2} b^2 + \sum_{i \in I} \mu_i (1 - \langle w, x_i \rangle - b) + \sum_{j \in J} \nu_j (1 - \langle w, y_j \rangle + b),$$

where $\mu_i \geq 0$ for all $i \in I$ and $\nu_j \geq 0$ for all $j \in J$. KKT conditions for the optimality of $\overline{w}$ and $B$ requires that

$$\overline{w} = \sum_{i \in I} \mu_i x_i + \sum_{j \in J} \nu_j y_j,$$

$$B = \sum_{i \in I} \mu_i - \sum_{j \in J} \nu_j,$$

and consequently, for each $k \in K$,

$$\langle \overline{w}, r_k \rangle = \sum_{i \in I} \mu_i \langle x_i, r_k \rangle - \sum_{j \in J} \nu_j \langle -y_j, r_k \rangle = \sum_{i \in I} \Delta_k \mu_i - \sum_{j \in J} \Delta_k \nu_j = B \Delta_k.$$

Then, we can write $\overline{w}$ as

$$\overline{w} = u + \sum_{k \in K} B \Delta_k r_k.$$

Remember, by definition,

$$w_{\text{SVM}} = \arg \min \|w\|^2 \ \text{s.t.} \ \langle w, x_i + y_j \rangle \geq 2 \quad \forall i \in I, \ \forall j \in J.$$

Since the vector $u$ also satisfies $\langle u, x_i + y_j \rangle = \langle w, x_i + y_j \rangle \geq 2$ for all $i \in I, j \in J$, we have $\|u\| \geq \|w_{\text{SVM}}\| = \frac{1}{\gamma}$. As a result, the margin obtained by minimizing the cross-entropy loss is

$$\frac{1}{\|\overline{w}\|} = \frac{1}{\sqrt{\|u\|^2 + \sum \|B \Delta_k r_k\|^2}} \leq \frac{1}{\sqrt{\frac{1}{\gamma^2} + B^2 \sum \Delta_k^2}}. \qquad \blacksquare$$

## D    PROOF OF THEOREM 4

If $B < 0$, we could consider the hyperplane $\langle \overline{w}, \cdot \rangle - B = 0$ for the points $\{-x_i\}_{i \in I}$ and $\{y_j\}_{j \in J}$, which would have the identical margin due to symmetry. Therefore, without loss of generality, assume $B \geq 0$. As in the proof of Theorem 3, KKT conditions for the optimality of $\overline{w}$ and $B$ requires

$$\overline{w} = \sum_{i \in I} \mu_i x_i + \sum_{j \in J} \nu_j y_j, \ \ B = \sum_{i \in I} \mu_i - \sum_{j \in J} \nu_j$$

where $\mu_i \geq 0$ and $\nu_j \geq 0$ for all $i \in I, j \in J$. Note that for each $k \in K$,

$$
\begin{aligned}
\langle \overline{w}, r_k \rangle &= \sum_{i \in I} \mu_i \langle x_i, r_k \rangle - \sum_{j \in J} \nu_j \langle -y_j, r_k \rangle \\
&= B\Delta_k + \sum_{i \in I} \mu_i(\langle x_i, r_k \rangle - \Delta_k) - \sum_{j \in J} \nu_j(\langle -y_j, r_k \rangle - \Delta_k) \ \geq \ B\Delta_k.
\end{aligned}
$$

Since $\{r_k\}_{k \in K}$ is an orthonormal set of vectors,

$$\|\overline{w}\|^2 \geq \sum_{k \in K} \langle \overline{w}, r_k \rangle^2 \geq \sum_{k \in K} B^2 \Delta_k^2.$$

The result follows from the fact that $\|\overline{w}\|^{-1}$ is an upper bound on the margin.    ∎

## E    PROOF OF THEOREM 6

In order to achieve zero training error in one iteration of the stochastic gradient algorithm, it is sufficient to have

$$\min_{i' \in I} \langle x_{i'}, x_i + y_j \rangle > \max_{j' \in J} \langle -y_{j'}, x_i + y_j \rangle \quad \forall i \in I, \ \forall j \in J,$$

or equivalently,

$$\langle x_{i'} + y_{j'}, x_i + y_j \rangle > 0 \quad \forall i, i' \in I, \ \forall j, j' \in J. \tag{9}$$

By definition of the margin, there exists a vector $w_{\text{SVM}} \in \mathbb{R}^d$ with unit norm which satisfies

$$2\gamma = \min_{i \in I, j \in J} \langle x_i + y_j, w_{\text{SVM}} \rangle.$$

Note that $w_{\text{SVM}}$ is orthogonal to the decision boundary given by the SVM. Then we can write every $x_i + y_j$ as

$$x_i + y_j = 2\gamma w_{\text{SVM}} + \delta_i^x + \delta_j^y,$$

where $\delta_i^x, \delta_j^y \in \mathbb{R}^d$ and $\|\delta_i^x\| \leq R_x$ and $\|\delta_j^y\| \leq R_y$. Then, condition (9) is satisfied if

$$\langle 2\gamma w_{\text{SVM}} + \delta_i^x + \delta_j^y, 2\gamma w_{\text{SVM}} + \delta_{i'}^x + \delta_{j'}^y \rangle > 0 \quad \forall i, i' \in I, \ \forall j, j' \in J,$$

or equivalently if

$$4\gamma^2 + 2\gamma \langle w_{\text{SVM}}, \delta_i^x + \delta_j^y + \delta_{i'}^x + \delta_{j'}^y \rangle + \langle \delta_i^x + \delta_j^y, \delta_{i'}^x + \delta_{j'}^y \rangle > 0 \quad \forall i, i' \in I, \ \forall j, j' \in J. \tag{10}$$

If we choose $\gamma > \frac{5}{2} \max(R_x, R_y)$, we have

$$4\gamma^2 - 2\gamma(2R_x + 2R_y) - (R_x + R_y)^2 > 0,$$

which guarantees (10) and completes the proof.    ∎

