# OpenReview forum: "Cross-Entropy Loss Leads To Poor Margins"
_ICLR.cc/2019/Conference_

### Official Review · AnonReviewer2 · 2018-11-02
**The technical results can be obtained by a simple combination of previous work.**

**Rating:** 5
**Confidence:** 4

**Review:**

Summary:
This paper investigates the properties of minimizing cross-entropy of linear functions over separable data (looks like logistic loss). The authors show a simple example where the minimizer of the cross-entropy loss leads to maximum margin hyperplane where the bias term is regarded as an extra dimension, which is different from the standard max. margin solution of  SVMs with bias not regarded as an extra dimension. The authors then propose a method to obtain the latter solution by minimizing the cross-entropy loss.


Comments:

There is a previously known result quite related to this paper:

Ishibashi, Hatano and Takeda: Online Learning of Approximate Maximum p-Norm Margin Classifiers with Bias, COLT2008.

Theorem 2 of Ishibashi et al. shows that the hard margin optimization with linear classifier with bias is equivalent to those without bias over pairs of positive and negative instances.

Combined with Theorem 3 of (Soudry et al., 2018)), I am afraid that the main result Theorem 5 can be readily derived.

For this reason, I am afraid that the main technical result is quite weak.

After Rebuttal:
I read the authors' comments. I understand more the technical contribution of the paper and raised my score. But I also agree with Reviewer 3.

---

> ### Author Response · Authors · 2018-11-07
> **Response to Reviewer 2: Main result is not Theorem 5**
>
> Dear Reviewer 2,
>
> Thank you for your review, and thanks for pointing out this reference. We were not aware of this past work, and it certainly deserves a reference.
>
> Nevertheless, our main technical result is Theorem 3 and Remark 3 -- not Theorem 5. As the title of our submission reflects, and as the list of our contributions on page 2 describes, differential training is not the heart of our work. As we stated in our response to Reviewer 1, differential training was introduced in this paper only to open a door for further research and not to finish this paper with a negative result.
>
> Please note that Theorem 3 and Theorem 4, along with the related remarks, are original. We would appreciate if you have any suggestions to further highlight that Section 3 is the critical part of our work.

---

> > ### Author Response · Authors · 2018-11-16
> > **Paper has been updated**
> >
> > 1) We changed the titles of Section 2 and Section 3 to reflect their importance.
> >
> > 2) We added citations to (Ishibashi et al., 2008) and one of its references, (Keerthi et al., 2000), in the first paragraph of Section 4.

---

### Official Review · AnonReviewer3 · 2018-11-03
**interesting work, but slightly incremental**

**Rating:** 5
**Confidence:** 4

**Review:**

This paper studies the cross-entropy loss for binary classification problems. The authors show that if the norms of samples in two linear separable classes are different, gradient descent based methods minimizing cross-entropy loss may give a linear classifier that gives small margin.

Pros

1. The paper is clearly written and very easy to follow.

2. The authors show that for two point classification problems, if the norms of the points are very different then gradient descent will give a very small margin.

3. Further theoretical results are given explaining the relation between cross-entropy loss and SVM.

4. A new loss function called differential training is proposed, which is guaranteed to give SVM solution.

Cons

1. My biggest concern is that, the paper, especially the title, may be slightly misleading in my opinion. Although the authors keep claiming that cross-entropy loss can lead to poor margins in certain circumstances (which I agree), in fact Theorem 1 and Theorem 2 have already clearly shown the connection between the cross-entropy solution and the maximum margin direction. For example, Theorem 1 literally proves that when the two points have the same norm (normalized data?), cross-entropy loss leads to maximum margin. Theorem 2 also clearly states that cross-entropy loss and SVM are closely related. Based on these two theorems, perhaps ‘cross-entropy loss is closely related to maximum margin’ is a more convincing statement.

2. The theoretical results given in this paper is slightly incremental. As the authors mentioned, Theorem 1 and Theorem 2 are essentially already proved in previous works. The other results are not very significant either.

3. The authors do not clearly state the advantages of the differential training method compared to SVM. It seems that one can just use SVM if the goal is maximum margin classifier.

---

> ### Author Response · Authors · 2018-11-07
> **Response to Reviewer 3**
>
> Dear Reviewer 3,
>
> Thanks for reviewing our paper.
>
> 1a) The goal of our submission is not to make further positive claims about the use of cross-entropy minimization; it is the opposite. As Reviewer 1 also stated, we wanted to challenge the faith of the community in the use of cross-entropy loss, and we wanted to show that minimizing this loss function on low-dimensional datasets such as images can lead to extremely poor margins. For this reason, the title of our submission is very accurate. We updated Figure 1 to highlight the drastic difference between the SVM solution and the solution obtained by the cross-entropy minimization.
>
> 1b) As we clearly stated in Remark 1, normalizing a dataset in the input space does not correspond to normalizing the features of the points if the feature mapping is nonlinear. In particular, we will not have normalized features if we use neural networks. If we want to get a right intuition about the effect of cross-entropy minimization on neural network training, we cannot simply assume the features of the training points will be normalized. This is why we strictly avoid the assumption of a normalized dataset, as explained in Remark 1.
>
> 2a) It is unfortunate, and somewhat curious, that our results in Section 3 (Theorem 3 and the remarks following it) were completely neglected. Section 3 clarifies why the conclusions of the works [1,2,3,4,5] are erroneous and shows that the reality is drastically different from their conclusions. Showing that there was a critical error in a line of previous works, which leads to a drastic change in the conclusion, is not an "incremental contribution". In fact, given [1] appeared in ICLR last year, it is essential that the ICLR community be given the correction this year.
>
> 2b) Theorem 3 and Remark 3 are the most critical results of our paper. Please make sure you have understood them. The last paragraph of Section 5 verifies that the assumptions of Theorem 3, the low-dimensionality of the features, indeed arises in practice. In other words, the assumptions of Theorem 3 are not an edge case, and the conclusion of Theorem 3 has critical implications for practice.
>
> 3) Our paper starts with the question "Is cross-entropy loss really the right cost function to use with gradient descent algorithm?". We use linear classifier and linearly separable dataset to answer this question on a simple setting. By doing so, our work gives intuition that the cross-entropy loss function has responsibility in the poor margin of the decision boundaries. We introduce differential training as a method to improve the margin **while still using gradient descent algorithm**. As we stated in the Discussion section, this allows the feature mapping to remain trainable while ensuring a large margin, and therefore, it provides an initial attempt to combine the benefits of neural networks and the SVM. And please note that when [1,2,3,4,5] claimed that cross-entropy loss finds the same solution with the SVM, they did not suggest that the ML community stop using cross-entropy minimization and replace it with SVM.
>
> [1] Daniel Soudry, Elad Hoffer, and Nathan Srebro. The implicit bias of gradient descent on separable data. In International Conference on Learning Representations, 2018.
> [2] D. Soudry, E. Hoffer, M. Shpigel Nacson, S. Gunasekar, and N. Srebro. The Implicit Bias of Gradient Descent on Separable Data. ArXiv e-prints, 2018.
> [3] M. Shpigel Nacson, J. Lee, S. Gunasekar, P. H. P. Savarese, N. Srebro, and D. Soudry. Convergence of Gradient Descent on Separable Data. ArXiv e-prints, 2018a.
> [4] M. Shpigel Nacson, N. Srebro, and D. Soudry. Stochastic Gradient Descent on Separable Data: Exact Convergence with a Fixed Learning Rate. ArXiv e-prints, 2018b.
> [5] Ziwei Ji and Matus Telgarsky. Risk and parameter convergence of logistic regression. CoRR, abs/1803.07300, 2018.

---

> > ### Comment · AnonReviewer3 · 2018-12-06
> > **Response**
> >
> > Sorry for my late reply! I've read the response, but I'm not convinced to change the rating.
> >
> > For 2a) and 2b), I apologize for not making my previous comment on Section 3 very clear. I did not ignore Theorems 3,4 and Remark 3. In my original comment, I tried to use 'Further theoretical results are given explaining the relation between cross-entropy loss and SVM.' to summarize these results. Because it seems that Theorems 3,4 further quantifies the relationship between margins given by cross-entropy and SVM based on Theorem 2. I'm still not convinced that these results are significant enough. Since the authors claimed that these are their main contributions, more explanation on the significance of these results should be added.
> >
> > For 1a), I'm still not convinced that it is appropriate to claim that the papers the authors mentioned are erroneous. It is very common that papers focusing on theoretical analysis make certain assumptions that do not exactly match what people do in practice. Normalizing the data and neglecting the bias terms can both be considered as such assumptions. When these assumptions are not satisfied, it is not surprising that most of the results won't hold. Also, as the authors and other reviewers have pointed out, Theorems 1,2 are already covered in existing works. Therefore, even if the papers the authors mentioned are indeed 'erroneous', it can hardly be considered as a contribution of this paper.
> >
> > For 1b), the authors argued that even if the data are normalized, the features of neural networks are still not normalized. This is true, but the current results on 'cross-entropy loss can lead to poor margins' are only shown for linear models. Without further results proving that neural networks with cross-entropy loss can give poor margins, it is still not very convincing.
> >
> > Because of the concerns above, I believe '5: Marginally below acceptance threshold' is an appropriate rating for this paper.

---

### Official Review · AnonReviewer1 · 2018-11-03
**A set of nice results that is insightful and clarifies some controversy**

**Rating:** 8
**Confidence:** 3

**Review:**

The paper challenges recent claims about cross-entropy loss attaining max margin when applied to linear classifier and linearly separable data. Along the road, it presents a couple of nice results that I find quite interesting and I believe they provide useful insights. Finally it presents a simple modification to the cross-entropy loss, which the authors refer to as differential training, that alleviates the problem for the case of linear model and linearly separable data.

CONS:
I find the paper useful and interesting mainly because of its insightful results rather than the final algorithm. The algorithm is evaluated in a very limited setting (linear model, synthetic data, binary classification); it is not clear if similar benefits would carry over to nonlinear models such as deep networks. In fact, I strongly encourage the authors to do a generalization comparison by comparing the **test accuracy** obtained by their modified cross-entropy against: 1. Vanilla cross-entropy as well as 2. A deep model large margin loss function (e.g. as in "Large Margin Deep Networks for Classification" by Elsayed). Of course on a realistic architecture and non-synthetic datasets (e.g. CIFAR-10).

PROS:
Putting the algorithm aside, I find the theorems interesting. In particular, Theorem 3 shows that some earlier claims about cross-entropy's ability to attain large margin (in the linearly separable case) is misleading (due to neglecting a bias term). This is important as it changes the faith of the community in cross-entropy and more importantly creates hope for constructing new loss functions with improved margin.
I also find the connection between the dimension of the subspace that contains the points and quality of margin obtained by cross-entropy insightful.

---

> ### Author Response · Authors · 2018-11-07
> **Response to Reviewer 1**
>
> Dear Reviewer 1,
>
> Thank you for reading our submission closely, and thanks for appreciating our results.
>
> As you have also noticed, Section 3 of our paper, and Theorem 3 in particular, is the punch line of our work. The algorithm, differential training, was introduced in this paper only to open a door for further research and not to finish this paper with a negative result. That is, we wanted to show that there could be a solution for the problem we have identified. We agree that further study of differential training for neural networks is necessary and important, and that is our ongoing work.

---

### Official Review · AnonReviewer5 · 2018-12-06
**Insufficient novelty and significance. Also, the phrasing of the results is somewhat misleading.**

**Rating:** 4
**Confidence:** 5

**Review:**

Due to the large variance in reviewer scores, I was asked to give this additional review.

Background: [Soudry et al. 2018] showed that the iterates of gradient descent, when optimizing logistic regression on separable data, converge to the L2 max-margin (SVM) solution for homogeneous linear separators (without bias). These results were later extended to other models and optimization methods.

This paper has two main results:
1)	It clarifies that the results of [Soudry et al. 2018] do not apply to logistic regression when the linear separator has a bias term (“b”). This is because the homogenous max margin solution in the extended [x,1] space is not the same as non-homogeneous max margin solution in the original space: the first has a penalty on the size of the bias term, i.e.
min_{w,b} ||w||^2 + b^2 s.t. y_n (w’x_n+b) >= 1
, while the latter does not:
min_{w,b} ||w||^2  s.t. y_n(w’x_n+b) >= 1
2)	It suggests using differential training to correct this issue.


However, I do not believe these contributions are enough for a publication in ICLR. First, (2) is simply a combination of two known results, as mentioned by Reviewer 2. Second, though I commend the authors for pointing out (1), I do not feel this by itself warrants a publication, for the following reasons:
a) It is very simple to explain (1) in only a few lines (as I did above). Therefore, it would be more informative just to write (1) as a comment on the original paper (the ICLR 2018 forum is still open), not as a completely new publication. For me, all the numerical demonstrations and examples of this simple issue did not add much.
b)	Regularizing the bias term usually does not make a significant difference to the sample complexity (see the end of section 15.1.1 in the textbook “Understanding Machine Learning: From Theory to Algorithms” by Shai Shalev Shwartz.). Furthermore, the main motivation behind [Soudry et al. 2018] was to explain implicit bias and generalization in deep networks, where there such max-margin results (which penalize all the parameters) could be used to derive generalization bounds (e.g., https://arxiv.org/abs/1810.05369).
c)	Lastly, the authors here say that “the solution obtained by cross-entropy minimization is different from the SVM solution”. This (as well as the title and abstract) may mislead the readers to think there is something wrong in the proofs of [Soudry et al. 2018] and later papers, and that logistic regression does not converge to the max-margin solution for homogeneous linear separators. However, the max-margin solution for homogeneous linear separators is also called the “max margin” or SVM solution (just for a different family). For example, see the previous paper on the topic [“Margin Maximizing Loss Functions”, Rosset et al. 2004] or section 15.1.1 in the textbook “Understanding Machine Learning: From Theory to Algorithms” by Shai Shalev Shwartz.  As I see it, the only issue in [Soudry et al. 2018] is the sentence “A bias term could be added in the usual way, extending x_n by an additional ’1’ component." which is confusing since it cannot be applied directly to the SVM solution. The authors should aim to pinpoint this issue, and clarify their phrasing to avoid such confusions.

---

> ### Author Response · Authors · 2018-12-06
> **Main results are Theorem 3-4 and Remark 3: They are completely, and probably intentionally, ignored**
>
> We repeatedly and clearly stated in our response to Reviewer 2 and Reviewer 3: Our most critical results are Theorem 3, Theorem 4 and Remark 3. Anyone who has read the list of our contributions on page 2 would not miss this. Anyone who has read the discussion section would understand that Theorem 3 is our most critical result -- just like Reviewer 1 did.
>
> We understand from the review that Reviewer 5 was able to see the previous reviews and our responses. Given this fact, Reviewer 5 must have seen in our responses that our most critical results are Theorem 3, Theorem 4 and Remark 3. Therefore, it seems extremely absurd that Reviewer 5 tried to summarize our contributions in two points and not mention any of our most critical results. As a result, we strongly question the objectivity and the fairness of their evaluation.
>
> Our paper is the first work that finds a connection between the existence of adversarial examples and the specific choice of training loss function (cross-entropy with soft-max) and the low dimensionality of the features of the training dataset. Anyone who thinks this result is insignificant should reconsider their level of expertise in the field and possibly give themselves confidence 1 or 2 -- not 5.
>
> We were very careful in our choice of words when making statements about what is correct and what is not correct in (Soudry et al., 2018). We stated **their conclusion was incorrect** due to neglecting the bias term; we did not say their proof was incorrect. Nevertheless, we appreciate the great effort Reviewer 5 did in praising the work (Soudry et al., 2018) and trying to remove the taint we could potentially bring to it while writing a review for our paper.

---

> > ### Comment · AnonReviewer5 · 2018-12-06
> > **Significance of Theorem 3-4 and Remark 3**
> >
> > The authors feel I ignored the results in Theorem 3,4 and remark 3. Specifically, as I understand, in these results the authors claim:
> > I)	From the mathematical perspective, one can find datasets (on near-affine subspaces) where the margin of the solution of cross-entropy minimization can be quite poor.
> > II)	From a practical perspective, neural networks tend to behave similarly to these examples and therefore have poor margins.
> >
> > I would like to clarify that I feel that “claim I” is again a simple demonstration of point (1) in my previous response (i.e. as I said “For me, all the numerical demonstrations and examples of this simple issue did not add much”). Specifically, since both the optimization problems in (1) are different, it feels clear to me that one can find examples where the solutions are very different, i.e. datasets where the in max margin solutions b>>||w||. It is straightforward to generate such datasets by strongly shifting the classes so a separator coming from the origin would have a poor margin, as done in Theorem 3. This is why I consider Theorem 3+4 as another example of (1).
> >
> > Moreover, I feel “claim II” is not sufficiently supported by evidence. Specifically, the authors demonstrate that the representation in CIFAR10 lies near an affine subspace as in theorem 3+4, but it is not clear if this B^2 sum_k \Delta_k^2 is indeed sufficiently large to hurt the margin. Remark 3 argues that this term B^2 sum_k \Delta_k^2 should be large in practice, in comparison to 1/gamma^2 but I don't see why this should be true, as both may scale with dimensions. To establish this claim, I think these quantities should have to been measured directly in the last layer of the network. The authors could have also directly measured the margin in the last layer and compared it to the max margin. Without these measurements, I do not feel that the authors indeed demonstrated "claim II".
> >
> > Lastly, I would like to clarify that in “issue c” in my previous response, I mainly wanted to point out to the authors that *some* of their phrasings were confusing (not all of them). For example, the statement “the solution obtained by cross-entropy minimization is different from the SVM solution” is wrong under some common interpretations (as SVM can also be defined for the class homogenous linear classifiers). Therefore, I feel they should be adjusted (“SVM”-> “SVM for linear predictors with bias”) to avoid further confusions.

---

### Official Review · AnonReviewer4 · 2018-12-06
**I do not think the proposed approach can be better than the cross-entropy loss in practice.**

**Rating:** 3
**Confidence:** 4

**Review:**

This paper presents a very specialized example to show that gradient descent on the cross-entropy loss WITHOUT REGULARIZATION leads to poor margin, which is very unrealistic. Moreover, I have the following concerns:

1. In the two points classification example shown in Section 2, I want to see the plot of iteration versus cross-entropy loss during the gradient descent.

2. Whether it makes sense to use cross-entropy loss to quantify loss for two-class classification problem with one point in each class? Statistically, it seems not reasonable at all.

3. In Corollary 1, the authors made a further assumption, x^Ty=1, which is very unnatural.

4. In the numerical results section, I want to see some results on some benchmark dataset. The presented numerical results are too weak to support the proposed differential training.

---

### Public Comment · ~Ignacio_Arroyo-Fernández1 · 2018-10-18
**Improve implementation of conditional nature of mutual information**

The proposed approach is very interesting as it revisits notions on margin-based and
discriminant classification, and brings those notions to model-based and Information-
Theoretic learning. However, there are two main concerns with the approach the authors
shoud seriously consider. The Cross entropy is a form of Mutual information, which is
in turn computed from two entropies. As this suggests, you have two probability measures
interacting. The events drawn from from one measure can occur given (or jointly
with) the occurrence of events drawn from the other. If the authors do not
consider these basic aspects, the convexity of the Mutual information measure can
be violated. Thus the measure does not converge. This is a possible cause for parameter
divergence. I suggets to consider these issues in order to improve the quality
of this paper, which provides a very interesting approach.

---

### Public Comment · ~Angus_Galloway1 · 2018-11-07
**weight decay baseline**

Could the authors compare their new loss function to *SGD* with cross entropy loss and (L2) weight decay with large regularization penalties?

---

### Meta-Review · Area_Chair1 · 2018-12-17
**Insufficient novelty**

**Confidence:** 4
**Recommendation:** Reject

**Metareview:**

The paper challenges claims about cross-entropy loss attaining max margin when applied to linear classifier and linearly separable data. This is important in moving forward with the development of better loss functions.

The main criticism of the paper is that the results are incremental and can be easily obtained from previous work.

The authors expressed certain concerns about the reviewing process. In the interest of dissipating any doubts, we collected two additional referee reports.

Although one referee is positive about the paper, four other referees agree that the paper is not strong enough.